# Polycaprolactone Films Modified by L-Arginine for Mesenchymal Stem Cell Cultivation

**DOI:** 10.3390/polym12051042

**Published:** 2020-05-02

**Authors:** Yuliya Nashchekina, Alina Chabina, Alexey Nashchekin, Natalia Mikhailova

**Affiliations:** 1Center of Cell Technologies, Institute of Cytology of the Russian Academy of Sciences, Tikhoretsky pr. 4, St. Petersburg 194064, Russia; chabina-alina@yandex.ru (A.C.); natmik@mail.ru (N.M.); 2Laboratory Materials and structures of Solid State electronics, Ioffe Institute, Polytekhnicheskaya str., 26, St. Petersburg 194021, Russia; nashchekin@mail.ioffe.ru

**Keywords:** polycaprolactone films, arginine, biofunctionalization, mesenchymal stem cells

## Abstract

This article describes the modification conditions and properties of polymer films obtained using a solution of poly(ε-caprolactone) modified with arginine. We investigated the effects on the surface and biological properties of films created using various arginine concentrations and temperature conditions during the modification process. We found that both increasing the arginine concentration of the treatment solution or the temperature of the treatment reaction increased the arginine content of the film. Following a cellular cultivation period of 3 days, greater levels of cell proliferation were observed on all modified poly(ε-caprolactone) films compared to unmodified polymer films. Experiments using fluorescence microscopy showed that the modification conditions also had a significant effect on cellular spreading and the organization of the actin cytoskeleton following 2 h of cultivation. The degree of spreading and actin cytoskeleton organization observed in cells on these modified polymer films was superior to that of the control films.

## 1. Introduction

Poly(ε-caprolactone) (PCL) is a member of the biodegradable polymers that belong to the polyester class. PCL is an aliphatic semi-crystalline polymer that, at physiological temperature, attains a rubbery state resulting in high toughness [1] and fine mechanical properties (high strength, elasticity). PCL is non-toxic and tissue-compatible, hence it has been widely used as a scaffold in regenerative medicine. The degradation time of PCL in tissues is from two to three years, depending on the molecular weight of the polymer and the shape and size of the scaffold. PCL scaffolds can be degraded either by microorganisms or through hydrolysis of their aliphatic ester bond under physiological conditions [2]. PCL contains five hydrophobic CH_2_ moieties in its repeating units that define its degradation time. PCL has the longest degradation time of all the polyesters, including polylactide, polyglycolide, and the copolymers.

PCL possesses hydrophobic properties and therefore does not meet the requirements for use as a manufactured scaffold (hydrophilicity is needed), thus its cellular compatibility must be enhanced for use in this area. At the date of publishing, it is known that the optimal water contact angle values for cell adhesion are in the range of 45–70 °C [3]. Currently, many attempts are being made to modify the surfaces of PCL scaffolds, including coating the surfaces with many different proteins, such as fibrin, fibronectin, gelatin, and collagen. Additionally, growth factors and proteoglycans have also been used for polymer modification. Reports have demonstrated that these modification methods result in finer initial cell adhesion, proliferation, and colonization. At the same time, these scaffold surface modifications, using proteins, have also demonstrated some disadvantages, as modifying the surface of the scaffold with foreign proteins alters the immune response following transplantation. Therefore, modification of the PCL surface using simple, single molecules is thought to be not only simpler, but also safer. Surface modification using the tripeptide arginine-glycine-aspartic acid (RGD) has been widely used [4,5]. The RGD tripeptide is part of the protein molecule responsible for cell adhesion to the extracellular matrix [6]. Binding of RGD to a scaffold surface requires the presence of a free functional group that is notably absent from PCL. In Zhang et al.’s works, PCL films were modified with 1,6-hexanediamine, followed by binding of the RGD peptide to a free amino group. It has been shown that RGD peptide significantly improves the biological properties of modified PCL films [7,8]. In the current literature, little attention has been paid to modifying polymer scaffolds with individual amino acids, despite the fact that modification with a single amino acid could not only improve the cell adhesion of scaffolds, but also simplify and reduce the manufacturing costs of the entire tissue-engineering structure [9]. One of the amino acids of a tripeptide is arginine.

Arginine is a dibasic amino acid that is a constituent of several proteins in the human body. Arginine is closely tied to several metabolic pathways involved in the synthesis of urea, nitric oxide (NO), polyamines, agmatine, and creatine phosphate [10,11]. The catabolism of arginine occurs via several enzymatic pathways. The two major arginine catabolic pathways occurring during wound healing are degradation via NO synthase (NOS) isoforms and the two arginase isoforms. The NOS predominates are typically released from macrophages. The resulting NO leads to the increased synthesis of collagen and angiogenesis. Therefore, arginine is an essential amino acid for wound healing [12]. However, despite the fact that arginine plays an important role in the process of tissue regeneration, few examples of its use in the modification of polymer scaffolds exist. Previous demonstrations have shown that polymer film modification with arginine decreases the film’s thrombogenicity by reducing the blood calcification rate [13,14].

However, despite the fact that arginine plays an important role in the process of tissue regeneration, few examples of its use in the modification of polymer scaffolds exist. It has been found that electrospun polyurethane modified with arginine counters oxidative stress in vitro, thus increasing its utility as a wound dressing [15]. Arginine has been used to modify PCL by electrospinning in the formation of scaffolds [16], and it was found that increasing the arginine concentration in the spinning solution decreased the diameter of the formed fibers. Concerning the cytotoxicity of arginine, it has been found that the optimal arginine concentration is 0.5–1 wt % in terms of cell adhesion and viability. Other research has demonstrated that the presence of arginine in PCL electrospun scaffolds influences the sample’s weight, wettability, water uptake, and hemocompatibility [17].

Mixing arginine with PCL solutions to form scaffolds through electrospinning has shown promising results, improving the biocompatibility and cytocompatibility of the formed scaffolds. However, using this modification technique, arginine becomes distributed throughout the entire volume of the scaffold, thus only a small quantity of arginine remains on the surfaces of the nanofibrils where it can directly interact with cells.

Using these previously described methods, it is impossible to control the amount of arginine on the surface of the modified polymer. In our study, we developed a method for the surface modification of PCL-based films. The controlled introduction of arginine molecules onto the polymer surface makes it possible to regulate the processes of cell adhesion and proliferation. Thus, the aim of this work was to develop the optimal conditions for arginine modification of PCL surfaces and to study the effects of differing amounts of arginine on the adhesion and proliferation of mesenchymal stromal cells (MSCs).

## 2. Materials and Methods

### 2.1. Film Formation

Polymers films were prepared using solvent techniques. PCL polymer powder (*M*_n_ = 80,000 g·mol^−1^; Sigma-Aldrich, St. Louis, MO, USA) was dissolved in chloroform (Reactiv, Saint-Petersburg, Russia). The polymer concentration in the solution was a 0.02 g /mL. The solution was then heated to 40 °C for solvent evaporation. The thickness of the PCL films was determined to be 5 μm. The PCL films were then sterilized using ozone treatment in order to study their cellular interactions. 

### 2.2. PCL Film Modification 

For the modification of PCL films, three different water solutions with varying concentrations of arginine were prepared, specifically 0.1, 0.25, and 0.5M. The PCL films were treated with the various arginine solutions at room temperature for either one or 24 h or at a temperature of 40 °C for 1 hr. Following the arginine treatment, the films were washed in water for 20 min and then dried.

### 2.3. Quantification of Arginine in the Modified Films

The amount of arginine bound to the PCL films was evaluated using the spectrophotometric method with the help of ninhydrin reaction. Previously, we developed a method for the quantification of amino acids in dimethyl sulfoxide (DMSO) that allows the user to obtain a homogeneous solution of PCL and amino acids [18]. To perform the ninhydrin reaction, the modified PCL films were treated with a solution consisting of 2.5 mL DMSO and 0.5 mL of 0.2% ninhydrin solution in DMSO. Matrix dissolution was performed for 15 min at a temperature of 100 °C. The solution was then cooled for 40 min at room temperature, and the absorbance of the reaction products was measured using a PE-5400UF spectrophotometer (ECOCHIM, Saint Petersburg, Russia) at a wavelength of 400 nm.

### 2.4. FTIR

Surface molecular structure of modified PCL samples was analyzed using a Fourier transform infrared (FTIR) IRPrestige-21 (Shimadzu, Tokyo, Japan) spectrometer, in absorbance mode, in 4000–600 cm^−1^ range. The resolution of measurement was 2 cm^−1^.

### 2.5. Scanning Electron Microscopy

The structure of the inner and outer layers of the polymeric vascular prosthesis was evaluated using a JSM-7001F (Jeol, Tokyo, Japan) scanning electron microscope (SEM). 

### 2.6. Cultivation of Mesenchymal Marrow Stromal Cells

MSCs were isolated from the flat pelvis bones of rabbits using a modified version of previously described methods [19]. For this purpose, the bone marrow was suspended in phosphate buffer solution (PBS), and then the suspension was layered onto Histopaque solution (Sigma, St. Louis, MO, USA) and centrifuged at 800 g for 20 min at room temperature. Bone marrow cells were cultivated in α-minimum essential medium (α-MEM; Lonza, St. Louis, MO, USA) supplemented with 10% fetal bovine serum (FBS; HyClone, St. Louis, MO, USA), 100 mg/mL streptomycin (Sigma-Aldrich, Steinheim, Germany), and 100 U/mL penicillin (Sigma-Aldrich, Steinheim, Germany). Cells were seeded in Petri dishes at a concentration of 1 × 10^6^ cells/cm^2^ and placed in a CO_2_-incubator with an atmosphere of 5% CO_2_ content at 37 °C. For our experiments, cells were used following 2–6 passages.

### 2.7. Adhesion and Proliferation of MSCs on Modified PCL Films

The adhesion and proliferation of MSCs on modified PCL films were analyzed by staining with gentian violet. Cells were seeded and cultivated on the modified films at 37 °C in a CO_2_ incubator for either 2 h, 1 day, or 3 days. Following the cultivation period, the medium was removed and the adherent cells were then washed with PBS and fixed using a 70% ethanol solution for 20 min. The MSCs were then stained with 0.1% gentian violet solution for 15 min. After staining, the films were washed with water and dried. To extract the dye from the cells, 100 µl of 7% acetic acid was added. The number of adhered and proliferated cells following the cultivation period was measured using photocolorimetric analysis, which is based on the absorbance of the dye (gentian violet) associated with cell proteins, using the Fluorofot “Charity” analyzer (SKB PROBANAUPRIBOR, St.-Petersburg, Russia). All measurements were made at a wavelength of 570 nm.

### 2.8. Fluorescence Staining of MSCs 

MSCs were fluorescence stained in order to study the effects of arginine modification on MSC adhesion, spreading, and the presence of focal contacts. In this experiment, pure glass was used as a positive control, while the negative control was unmodified PCL film.

Cells were seeded and cultivated on the modified films at 37 °C in a CO_2_ incubator for either 2 h or 1 day. After the cultivation period, the medium was removed and the adherent cells were washed with PBS, fixed with a 4% formaldehyde solution (Sigma-Aldrich, Saint Louis, MO, USA) for 10 min, and then washed three times with PBS. Next, a detergent solution consisting of 0.1% Triton X-100 (Sigma-Aldrich, Saint Louis, MO, USA) was added to the cells for 15 min and then washed off with PBS. Preparations were then stained with rhodamine phalloidin (Thermo Fisher Scientific, Carlsbad, CA, USA) for 15 min, in order to stain the cytoskeleton, and then washed with PBS. Finally, the cells were treated with a mounting-medium containing DAPI (ab104139; Abcam, Cambridge, MA, USA). The actin cytoskeleton organization was then observed using a confocal microscope Olympus FV3000 (Olympus Corporation, Tokyo, Japan).

To study the presence of focal contacts using MSCs cultivated on the modified PCL films, the cells were incubated with anti-vinculin antibodies. Following 1 day of cultivation, the culture medium was removed and the cells on modified films were washed with PBS. The cells were then fixed with a formalin solution for 10 min and then washed three times with a 0.1% Tween PBS solution. Next, the cells were treated with 0.1% Triton X-100 for 15 min and then washed again three times with 0.1% Tween PBS solution. After the final washing process, anti-vinculin rabbit antibodies (ab129002; Abcam, Cambridge, MA, USA) were added to the cells for a 24 hr period. Following this incubation, the polymer films were carefully washed with PBS and then goat anti-rabbit IgG (H & L chain) antibodies (ab205718; Abcam, Cambridge, MA, USA) were added for 45 min. Next, the cells were washed with PBS containing 0.1% Tween and then treated with mounting medium containing DAPI. Then, using a confocal microscope Olympus FV3000 (Olympus Corporation, Tokyo, Japan), we observed the cells for the presence of focal contacts.

### 2.9. Cell and Focal Contacts Counts

To study the effects of the modified PCL films on cellular adhesion and proliferation, cells were cultured for a period of either 2 h, 1 day, or 3 days, and then the cells were counted. For this purpose, five different pictures of fields on each matrix were taken at a wavelength of 365 nm (DAPI) using a fluorescence microscope Pascal (Carl Zeiss Jena GmbH, Jena, Germany). The ImageJ program was used to count the nuclei in each picture and cell counts were based on the number of colored cell nuclei in the pictures. The ImageJ program was also used to count the number of focal contacts [20].

### 2.10. Statistical Analysis

All experiments were performed in 3–5 replicates. A T-test was performed using Microsoft Excel software to analyze the statistically significant differences between specific samples. Samples were considered to be statistically important with the p < 0.05.

## 3. Results and Discussion

The incorporation of arginine onto the PCL scaffolds was successfully achieved via direct addition of arginine into a polymeric solution [4,5]. Modification of polymer scaffolds with arginine has repeatedly been shown to increase the biocompatibility and cytocompatibility of the synthetic polymer scaffolds. In our study, we developed a method for the surface modification of PCL films. The results of the ninhydrin reaction demonstrated that enhancing the concentration of arginine in the polymer solution, from 0.1 to 0.5 M, increased the amount of bound arginine (Figure 1). Increasing the processing temperature also contributed to an increase in the amount of bound arginine. Our preliminary results showed that the processing time also affected the amount of bound arginine. In our experiments, maximal amounts of arginine were bound to the PCL surface following either treatment of the films with the arginine solution at room temperature for 1 day or treatment for 1 h at 40 °C. A longer treatment time of the films at room temperature did not increase the amount of bound arginine. Increasing the treatment time with arginine solution under heated conditions (40 °C) led to destruction of the films. From literature data, it is known that the hydrolysis of PCL is maximal at 1 h of water solution incubation at 40 °C [4]. The degradation process of the polymer ester bonds actively starts at this temperature. As a result, bonds are broken and free carboxyl–COOH and hydroxyl –OH groups are formed on the PCL film surface. In these conditions, the arginine amino groups in the solution begin to interact with the free PCL carboxyl groups. The results of the ninhydrin reaction showed that, at the same concentration of arginine solution but different temperatures, the amount of bound arginine was significantly different (Figure 1). In other words, increasing the treatment temperature from room temperature to 40 °C led to an increase in the amount of arginine bound to the polymer by a factor of three. We assume these results are due to the mechanism of the aminolysis reaction. PCL molecules have abundant ester groups (–COO–) that can be hydrolyzed to carboxylic acid under hydrolysis. Arginine is an amino acid that contains two amino groups, and it is possible that the amino groups can be introduced onto the polyester surface by a reaction with diamine, providing that one amino group reacts with the –COO– group to form a covalent bond, –CONH–, while the other amino group remains unreacted and is free.

The surface molecular structure of the modified PCL samples was analyzed using FTIR spectrometer. Figure 2 shows the FTIR spectra of PCL films before and after the arginine treatment. There are two amide bands resulting from a nucleophilic attack of arginine on the carbonyl group of PCL, which may be treated as evidence of an arginine interaction with the PCL surface. The first band in the range 1510–1580 cm^−1^ corresponds to amide II and is associated with N–H bending vibrations. The second band in the range 1600–1700 cm^−1^ is assumed to correspond to amide I and be associated with C=O (carbonyl) stretching vibration and C–N group vibrations. The resolving power of the FTIR spectrometer revealed the amide bond only in PCL samples treated with a 0.5 M arginine solution at a temperature of 40 °C. The results of the ninhydrin analysis demonstrated that with such parameters of modification, most of the arginine binds to the surface. At room temperature, signals corresponding to the amide bond in the above ranges were not detected. 

Aminolysis treatment represents an easy-to-perform chemical technique and has been extensively employed in activating biodegradable synthetic scaffolds [21,22]. Ethylenediamine and 1,6-hexanediamine are the main diamines that are used to process polymers. Our results demonstrating that the amount of arginine bound to the polymer surface depends on the solution concentration, temperature, and processing time are consistent with previously obtained data from another research group that used the aminolysis reaction to create PCL films treated with other diamine [9].

Isopropyl alcohol was usually used as a solvent for diamine for the modification of PCL films. In the work of Zhang and Hollister, 1,6-hexanediamine was dissolved in isopropyl alcohol and used to modify PCL films [7]. The modification was carried out at 37 °C. A larger number of adherent cells was observed on samples modified with RGD peptide compared to cells cultivated on diamine-treated PCL films. However, the presence of isopropyl alcohol during the processing of PCL films can lead to its partial sorption on the film, which will subsequently affect cell viability. In our study, the films were treated with an aqueous solution of arginine.

The changes in topography of the PCL film surfaces after modification were characterized using scanning electron microscopy (SEM). Figure 3 shows the representative SEM images of the unmodified PCL and arginine solution treatment PCL films. The initial state of the PCL surface had a relatively smooth surface. PCL is a crystalline polymer with a low melting point [23]. During the film fabrication, irregular hexagonal cells were formed. The size and distribution of these cells is shown in Figure 4. The cell size does not exceed 100 µm. The SEM results do not reveal significant changes in the topology of the PCL surface after modification. Neither the concentration of the arginine solution, the processing time or the temperature influenced the surface topology. The absence of pores on the PCL surface allows us to conclude that the physical sorption of arginine makes an insignificant contribution to the total amount of arginine on the modified film.

Other researchers have shown that, when using arginine injection into PCL solution during the preparation of scaffolds using the electrospinning method, arginine affects the diameter of the formed fibrils [16]. The conditions of our modification did not reveal significant changes in the surface topology of the PCL films before and after treatment with arginine solution.

Cellular adhesion plays an integral role in cell regulation and communication and is of fundamental importance in the development and maintenance of tissues. It is important to understand how cells interact and coordinate their behaviors on polymer surfaces. According to the “cell adhesion model”, the more a cell sticks, the greater the number of chemical bonds it has on its surface [24,25]. Cell adhesion is involved in stimulating signals that regulate the cell cycle, cell differentiation, cell migration, and cell survival [26]. A cell’s affinity to its substrate is a crucial consideration in biomaterial design, manufacturing, and development [27].

We studied cellular adhesion and proliferation following the cultivation periods of 1 and 3 days by staining cells with gentian violet. This dye stains living cells and allows the user to evaluate cell viability on modified films. In analyzing our films, a plain glass surface was used as the positive control, thus the number of adhered and proliferating cells on the glass surface was taken to be 100%. Our results demonstrate that modifying the PCL films with arginine significantly increased the number of adhered and proliferating cells (Figure 4). It should be noted that, after MSC cultivation for 1 day on films modified with a 0.5 M arginine solution at 40 °C, the number of adherent cells was less than that of other samples, including the unmodified PCL films. The previous modified arginine sample corresponds to a maximum of 0.2 μg per 1 cm^2^ of arginine on the polymer film. Based on these results, it can be assumed that too much arginine adversely affects MSC adhesion. It should also be noted that, in parallel with the aminolysis reaction under these conditions, partial hydrolysis of the PCL film surface also occurs. As a result of the hydrolysis, ester bonds are destroyed and negatively charged carboxyl groups are formed. Reviewing the data in the currently available literature, it is known that surface chemistry and charge character are also very important factors in terms of cellular adhesion. It seems that carboxylic acid groups grafted on the polymer surface had a negative effect on cell adhesion, spreading, and growth, which is consistent with previous suggestions by others researchers [28,29]. 

Maximal numbers of adherent cells were observed on the films treated with 0.25 M arginine solution at 40 °C and 0.5 M arginine solution at room temperature. We assume that these precise conditions contribute to a greater degree of aminolysis reactions compared to hydrolysis reactions and, consequently, the formation of a larger number of positively charged amino groups on the PCL surface compared to negatively charged carboxyl groups.

Following a cell cultivation period of 3 days, the proliferation of MSCs on all modified PCL films was greater than that on unmodified PCL films. Specifically, treating the films with arginine solution at 40 °C led to a two-fold increase in cell proliferation compared to MSCs cultivated on unmodified PCL surfaces. Based on these results, it can be concluded that all processing conditions contribute to the proliferation of MSCs. 

Our results are in line with previously obtained results of other researchers [7]. During the first days after seeding, the number of adherent MSCs on modified PCL films is less than on an untreated sample. After 3 days cultivation, the number of MSCs on untreated films after all modifications is greater than on pristine films.

The results displayed in Figure 5 suggest that surface chemistry and charge character are important factors determining cellular adhesion and spreading. Cells transmit extracellular and intracellular forces through localized sites where they are adhered to surfaces. The adhesion sites are formed by transmembrane proteins called integrins that anchor the cell to a polymer surface [30]. The integrins are attached to the tensile members of the cytoskeleton, actin filaments, through the focal adhesion (FA) complex, which is a highly organized cluster of molecules [31]. The cytoskeletal structure holds the nucleus in place and maintains the shape of the cell [32]. Integrins play an important role in mechanotransduction through FA proteins that link the integrin domains to the actin filaments. The FA formation is important in cell signaling in order to direct cell migration [33], proliferation, and differentiation [34] for tissue organization, maintenance, and repair.

The results of our fluorescence microscopy experiments showed that, after 2 h of cultivation, the conditions under which films are modified has a significant effect on cellular spreading and actin cytoskeleton organization. Figure 3 shows that the greatest degree of cell spreading was observed on PCL films treated with arginine solution at room temperature. The degree of spreading and actin cytoskeletal organization of MSCs on these films was superior to that of the control plates. There was no significantly different effect on the number of cells adhered to film samples treated with arginine solution at 40 °C compared to room temperature.

Following a cultivation period of 1 day, there was no significant difference in the organization of the MSC actin cytoskeleton between the samples (Figure 6). MSCs with well-developed actin cytoskeleton structures and well-organized actin filaments were observed on all samples. The results from the quantitative analysis using fluorescence microscopy were consistent with the results from the cell counting using the gentian violet method.

The results of staining cells with antibodies to vinculin, a focal contact protein, confirmed the previous results. After 1 day of cultivation, modification of the PCL films with arginine solution, either under heat or at room temperature, promoted cellular spreading and the formation of focal contacts. It should be noted that the cell spreading is correlated with the amount of arginine on PCL surface. The more arginine bound to PCL (Figure 1), the larger the spreading cells (Figure 7d). MSCs spread more on treated PCL surfaces compare to the control (both positive and negative). The number of focal contacts after cultivation for 1 day does not directly depend on the amount of bound arginine. We observed the largest number of focal contacts in MSCs cultured on PCL films treated with 0.5 M arginine solution at room temperature. While on the PCL sample with the largest amount of arginine treated with a 0.5 M arginine solution at heating, we did not find focal contacts (Figure 7f). We can assume that during the first days of cultivation, the number of focal contacts may change over time. Thus, it was previously shown that after 8 and 24 h not only the number of focal contacts, but also their localization depends on the properties of PCL-based surfaces [35]. We can also assume that too much arginine on a PCL surface prevents the formation of focal contacts, but we will investigate this assumption in the next research.

The results of fluorescence microscopy showed that, after a cultivation period of 2 h, the greatest degree of spreading was observed in the cells cultured on PCL films modified with arginine solution at room temperature, while the cells on films that were heated during arginine modification were more spheroid. As noted earlier, the surface charge is an important factor in determining cellular adhesion to a surface. Specifically, cells adhere better to a positively charged surface than to a neutral one. Thus, we assume that the largest positive charge exists on the surfaces of films modified with arginine solution at room temperature. When the arginine solution is heated on the surface of the films, hydrolysis reactions occur, which results in the formation of negatively charged carboxyl groups that neutralize the positively charged amino groups of the arginine. Therefore, cells on heated films are less flattened than cells on films modified with arginine solution at room temperature. After 1 day of cultivation, the effect of surface charge on cell adhesion is reduced since, during the cultivation period, nutrient medium components, including serum proteins that play an important role in cell adhesion, are adsorbed onto the polymer surface.

## 4. Conclusions

Arginine concentration and treatment temperature affect on the amount of PCL-bound arginine. It was shown that the maximum amount of arginine bond with the polymer surface occurred during modification with a 0.5 M arginine solution at a temperature of 40 °C. Polycaprolactone films modified via the addition of arginine were successfully produced for MSC cultivation. The presence of arginine influenced the degree of cellular adhesion, spreading, and proliferation. Additional studies are necessary to investigate the ideal in vitro conditions for cell cultivation and for structural and mechanical changes to occur in the films during their processing with arginine solution.

## Figures and Tables

**Figure 1 polymers-12-01042-f001:**
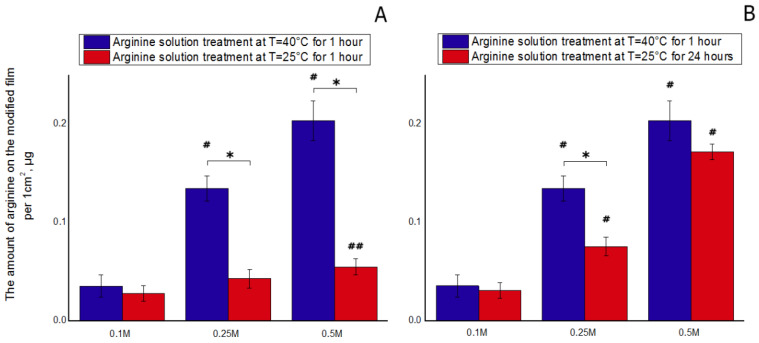
The dependence of the amount of arginine on the modified film per 1 cm^2^ on the concentration of the solution at T = 40 °C and T = 25 °C: (**A**) with the same duration of treatment, (**B**) with the maximum duration of treatments (n = 5: *—*p* < 0.01 for the same concentration data, #—*p* < 0.01 compared with lower concentration, but the same temperature, ##—*p* < 0.05 compared with 0.1 M at 25 °C).

**Figure 2 polymers-12-01042-f002:**
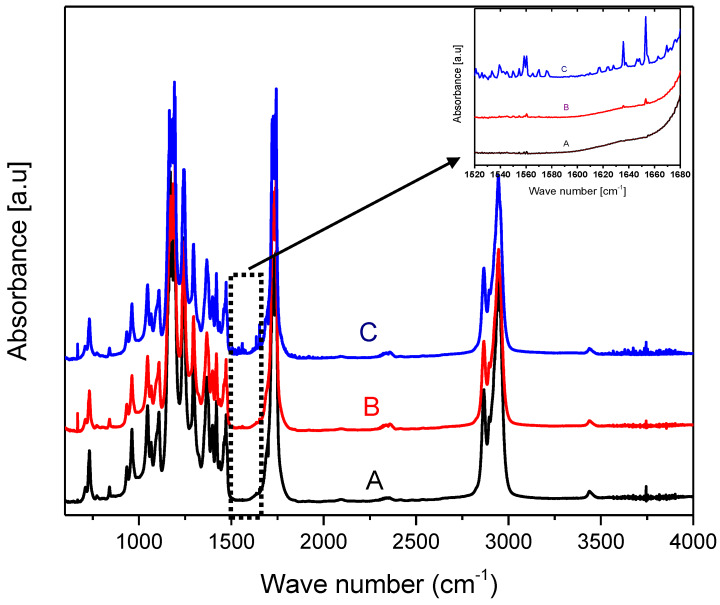
Fourier transform infrared (FTIR) spectra of the various films: (A) unmodified poly(ε-caprolactone) (PCL), (B) PCL treated with the 0.1 M arginine solution for 24 h at T = 25 °C, (C) PCL treated with the 0.5 M arginine solution for 1 h at T = 40 °C. The two additional peaks at the range from 1550 to 1650 cm^−1^ in aminolysed PCL films (C) indicated the introduction of amine groups onto the PCL substrates.

**Figure 3 polymers-12-01042-f003:**
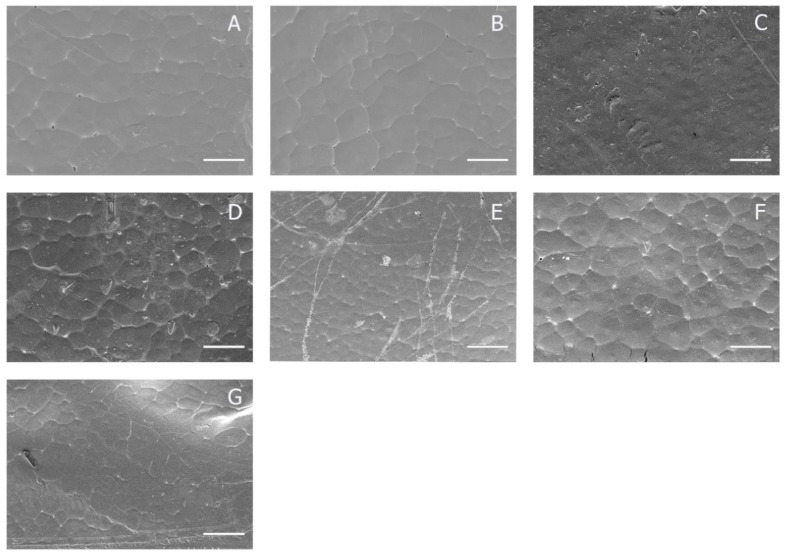
Scanning electron microscopy (SEM) images of PCL films; (**A**) unmodified PCL film; (**B**) film treated with the 0.1 M arginine solution for 1 h at T = 40 °C; (**C**) film treated with the 0.25 M arginine solution for 1 h at T = 40 °C; (**D**) film treated with the 0.5 M arginine solution for 1 h at T = 40 °C; (**E**) film treated with the 0.1 M arginine solution for 24 h at T = 25 °C; (**F**) film treated with the 0.25 M arginine solution for 24 h at T = 25 °C; (**G**) film treated with the 0.5 M arginine solution for 24 h at T = 25 °C. Scale bar 100 µm.

**Figure 4 polymers-12-01042-f004:**
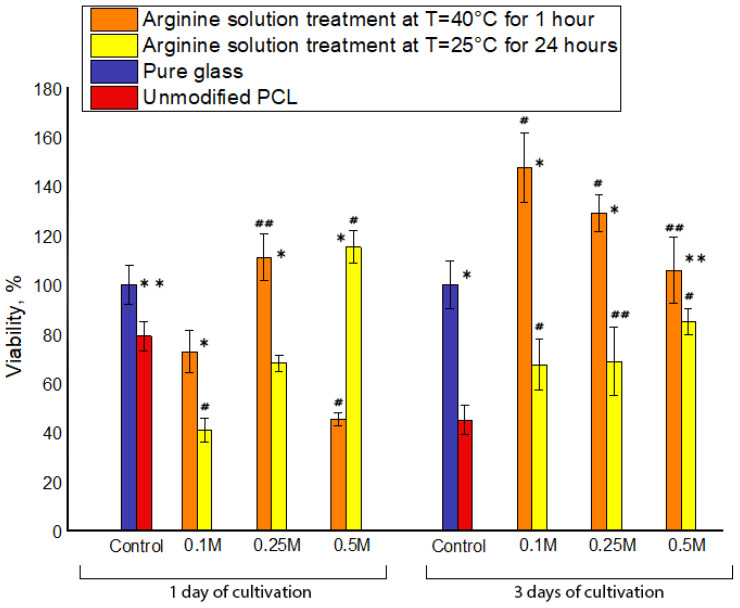
The dependence of viability of the MSC cells on the type of treatment of PCL films and the duration of cultivation (n = 5: *—*p* < 0.01, **—*p* < 0.05 for the same concentration data, #—*p* < 0.01, ##—*p* < 0.05 compared with the unmodified PCL).

**Figure 5 polymers-12-01042-f005:**
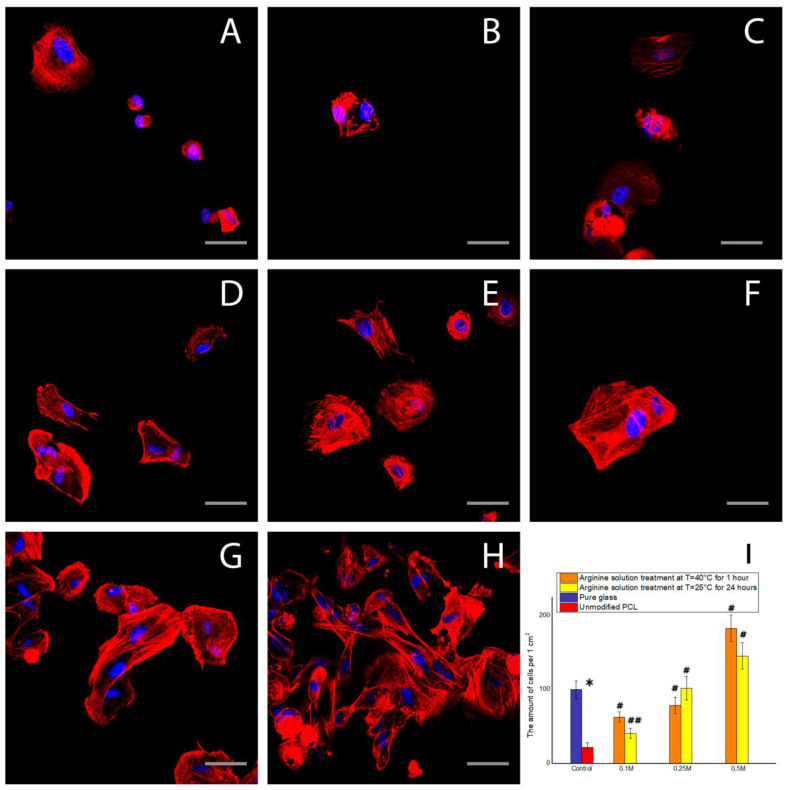
Fluorescence microscopy of the MSC cells after 2 h of cultivation: (**A**) on the pure glass, (**B**) on the unmodified PCL, (**C**) on the film treated with the 0.1 M arginine solution for 1 h at T = 40 °C, (**D**) on the film treated with the 0.25 M arginine solution for 1 h at T = 40 °C, (**E**) on the film treated with the 0.5 M arginine solution for 1 h at T = 40 °C, (**F**) on the film treated with the 0.1 M arginine solution for 24 h at T = 25 °C, (**G**) on the film treated with the 0.25 M arginine solution for 24 h at T = 25 °C, (**H**) on the film treated with the 0.5 M arginine solution for 24 h at T = 25 °C, and (**I**) the dependence of the amount of MSC cells on the modification conditions after 2 h of cultivation (n = 5: *—*p* < 0.01 for the same concentration data, #—*p* < 0.01, ##—*p* < 0.05 compared with the unmodified PCL). Scale bar 50 µm.

**Figure 6 polymers-12-01042-f006:**
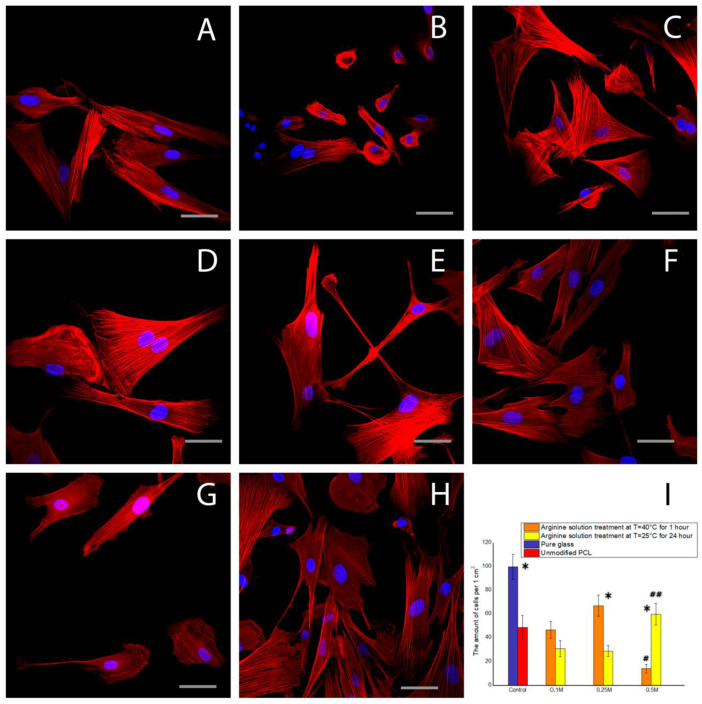
Fluorescence microscopy of the MSC cells after 1 day of cultivation: (**A**) on the pure glass, (**B**) on the unmodified PCL, (**C**) on the film treated with the 0.1 M arginine solution for 1 h at T = 40 °C, (**D**) on the film treated with the 0.25 M arginine solution for 1 h at T = 40 °C, (**E**) on the film treated with the 0.5 M arginine solution for 1 h at T = 40 °C, (**F**) on the film treated with the 0.1 M arginine solution for 24 h at T = 25 °C, (**G**) on the film treated with the 0.25 M arginine solution for 24 h at T = 25 °C, (**H**) on the film treated with the 0.5 M arginine solution for 24 h at T = 25 °C, and (**I**) the dependence of the amount of the MSC cells on the modification conditions after 1 day of cultivation (n = 5: *—*p* < 0.01 for the same concentration data, #—*p* < 0.01, ##—*p* < 0.05 compared with the unmodified PCL). Scale bar 50 µm.

**Figure 7 polymers-12-01042-f007:**
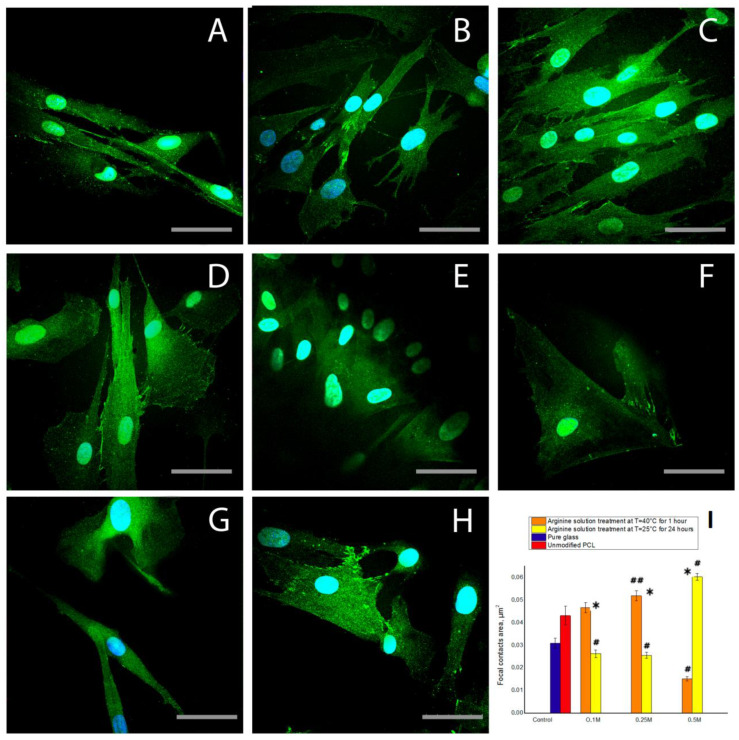
Presence of focal contacts of the MSC cells after 1 day of cultivation: (**A**) on the pure glass, (**B**) on the unmodified PCL, (**C**) on the film treated with the 0.1 M arginine solution for 1 h at T = 40 °C, (**D**) on the film treated with the 0.25 M arginine solution for 1 h at T = 40 °C, (**E**) on the film treated with the 0.5 M arginine solution for 1 h at T = 40 °C, (**F**) on the film treated with the 0.1 M arginine solution for 24 h at T = 25 °C, (**G**) on the film treated with the 0.25 M arginine solution for 24 h at T = 25 °C, (**H**) on the film treated with the 0.5 M arginine solution for 24 h at T = 25 °C, and (**I**) the dependence of the focal contacts area on the modification conditions after 1 day of cultivation (n = 5: *—*p* < 0.01 for the same concentration data, #—*p* < 0.01, ##—*p* < 0.05 compared with the unmodified PCL). Scale bar 50 µm.

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
