# Peer review of "Polycaprolactone Films Modified by L-Arginine for Mesenchymal Stem Cell Cultivation"

_polymers, 2020, doi:10.3390/polym12051042_

Round 1
Reviewer 1 Report
The article describes about poly caprolactone (PCL) modified arginine for better mesenchymal stem cell attachment. Similar to this research, Goreninskii et al (reference no. 12) have developed arginine-doped PCL electrospun composite scaffolds and studied for mesenchymal stem cells viability and adhesion. In this study, author claims that arginine can be responsible for better cell adhesion, and author produced PCL films and modified with arginine. Further, cell culture studies were performed and found that arginine solution treatment at 40 °C for 1 hour shown better attachment compared to other controls. Although, the amount of results produced in the study very limited, but can be acceptable for publication.
Author Response
Response
Dear Reviewer 1, we are very grateful to you for manuscript review and valuable comments. We tried to take into account all your comments and recommendations. We added some new results on the modified surface of a PCL film by FTIR and SEM methods. The resolution of the FTIR spectrometer let us to confirm the amide bond only in PCL film treated with 0.5 M arginine solution at heating. SEM results showed that treatment with arginine solution does not affect the surface topology of PCL films. The results were also supplemented by comparison with data of other researchers. We hope that we have improved the manuscript and took into account your comments. Thanks a lot.

Reviewer 2 Report
The manuscript describes the modification conditions and properties of polymer films obtained using a solution of PCL modified with L-arginine. In fact, surface modification of PCL structures remains a key research topic.
The manuscript is original containing information that needs to be complemented. It demonstrates a reasonable understanding of the relevant literature in the field, however, the range of literature sources reported/reviewed need to be improved. The methods are appropriate employed. Results and Discussion as well as Conclusion need to be improved.
Major revision:
Authors should add the following references (at least):
- REF1 - Huina Zhang and Scott Hollister. “Comparison of Bone Marrow Stromal Cell Behaviors on Poly(caprolactone) with or without Surface Modification: Studies on Cell Adhesion, Survival and Proliferation”, Journal of Biomaterials Science 20 (2009) 1975–1993.
- REF2 - Huina Zhang, Chia-Ying Lin, Scott J. Hollister. “The interaction between bone marrow stromal cells and RGD-modified three-dimensional porous polycaprolactone scaffolds”, Biomaterials 30 (2009) 4063–4069.
Results and Discussion:
- Zhang et al. 2009 (REF1) studied the PCL film surface modified with RGDC peptide by a chemical immobilization procedure. Thus, authors should provide a clear comparison/explanation (advantages/drawbacks…) between the proposed manuscript and the Zhang’s paper.
- In the Goreninskii’s paper (Ref. 12 in the actual manuscript), composite fibrous scaffolds were successful obtained by electrospinning of a solution of PCL and L-arginine in hexafluoro-2-propanol. The influence of L-arginine content on structure, mechanical, surface and biological properties of the scaffolds was investigated. Thus, the authors aAe invited to provide a more in-deep discussion between the Goreninskii’s paper, the paper mentioned in REF2 and the proposed manuscript (advantages/drawbacks, applications)
- The authors mention that “additional studies are necessary to investigate the ideal in vitro conditions for cell cultivation and for structural and mechanical changes to occur in the films during their processing with arginine solution”. In fact, these studies are relevant and it will enhance the manuscript. A study on mechanical properties is encouraged to be included in the actual manuscript.
- The temperature of 37ºC is a physiological condition and it is near to 40ºC. Why not 37ºC?
Minor issues:
Please, provide the references in the correct format (see line 336 and others).
Author Response
The manuscript describes the modification conditions and properties of polymer films obtained using a solution of PCL modified with L-arginine. In fact, surface modification of PCL structures remains a key research topic.
The manuscript is original containing information that needs to be complemented. It demonstrates a reasonable understanding of the relevant literature in the field, however, the range of literature sources reported/reviewed need to be improved. The methods are appropriate employed. Results and Discussion as well as Conclusion need to be improved.
Dear Reviewer 2, we are very grateful to you for manuscript review and valuable comments. We added some new results and have improved the manuscript, took into account your comments and recommendations. Thanks a lot. .
Major revision:
Authors should add the following references (at least):
- REF1 - Huina Zhang and Scott Hollister. “Comparison of Bone Marrow Stromal Cell Behaviors on Poly(caprolactone) with or without Surface Modification: Studies on Cell Adhesion, Survival and Proliferation”, Journal of Biomaterials Science 20 (2009) 1975–1993.
- REF2 - Huina Zhang, Chia-Ying Lin, Scott J. Hollister. “The interaction between bone marrow stromal cells and RGD-modified three-dimensional porous polycaprolactone scaffolds”, Biomaterials 30 (2009) 4063–4069.
We are very grateful to your valuable comments and useful reference. Sorry, that we did not include them in the manuscript at the beginning, they are very helpful to us. These articles were studied in detail and a comparative analysis of our results with published data was carried out. The main distinguishing feature of our research is the use of arginine as a source of diamine. Arginine is one of the components of the RGD peptide. So far, we can only assume that arginine is more cytocompatible than synthetic diamines.
Results and Discussion:
- Zhang et al. 2009 (REF1) studied the PCL film surface modified with RGDC peptide by a chemical immobilization procedure. Thus, authors should provide a clear comparison/explanation (advantages/drawbacks…) between the proposed manuscript and the Zhang’s paper.
We agree with the reviewer's comments and performed comparative analysis between modification conditions and interaction of cells on modified films. Indeed, modification of the RGD peptide significantly improves the properties of the PCL films. However, this method requires the use of chemical agents and they can affect on cells and surrounding tissues after polymer degradation.
-In the Goreninskii’s paper (Ref. 12 in the actual manuscript), composite fibrous scaffolds were successful obtained by electrospinning of a solution of PCL and L-arginine in hexafluoro-2-propanol. The influence of L-arginine content on structure, mechanical, surface and biological properties of the scaffolds was investigated. Thus, the authors aAe invited to provide a more in-deep discussion between the Goreninskii’s paper, the paper mentioned in REF2 and the proposed manuscript (advantages/drawbacks, applications)
It was shown that the arginine injection into PCL solution during the preparation of scaffolds by the electrospinning method, arginine effects on the formed fibrils diameter [Goreninskii et al., 2018]. The conditions of our modification did not reveal significant changes in the surface topology of the PCL films before and after treatment with arginine solution.
In the Goreninskii’s paper the structural and mechanical characteristics of the obtained scaffolds are described in sufficient detail. The PCL solution consists of various arginine content during scaffolds formation. However, there is no data on the quantitative content of arginine after process of scaffolds formation. This does not allow us conclude directly that the cells interaction depend of the amount of arginine in the scaffold.
S.I. Goreninskii, E.N. Bolbasov, E.A. Sudarev, K.S. Stankevich, Y.G. Anissimov, A.S. Golovkin, A.I. Mishanin, A.N. Viknianshchuk, V.D. Filimonov, S.I. Tverdokhlebov Fabrication and properties of L-arginine-doped PCL electrospun composite scaffolds. Mat. Lett. 2018, 1, 64-67.
-The authors mention that “additional studies are necessary to investigate the ideal in vitro conditions for cell cultivation and for structural and mechanical changes to occur in the films during their processing with arginine solution”. In fact, these studies are relevant and it will enhance the manuscript. A study on mechanical properties is encouraged to be included in the actual manuscript.
We are grateful to the reviewer for this valuable recommendation. Indeed, we have done research to evaluate the mechanical characteristics of the PCL films. A lot of results have been obtained and we are currently preparing next manuscript. This manuscript describes the various structural and mechanical characteristics of the PCL samples. Some of these results are listed below. But we decided not to put them in this manuscript so as not to increase the volume of manuscript and publish separately.
. - The temperature of 37ºC is a physiological condition and it is near to 40ºC. Why not 37ºC?
From pubished date it is known that the hydrolysis of PCL was maximal when incubated at 40°C in water solution during 1 h [Chua, et al., 2013]. The degradation process of the polymer ester bonds actively starts at this temperature. As a result, bonds are broken and free carboxyl - COOH and hydroxyl -OH groups are formed on the PCL film surface. At these conditions the arginine amino groups begin to interact with the free PCL carboxyl groups in the solution.
T.-K. Chua, M. Tseng, M.-K. Yang Degradation of Poly(ε-caprolactone) by thermophilic Streptomyces thermoviolaceus subsp. Thermoviolaceus 76T-2 AMB Express 3 (2013) 8
Minor issues:
Please, provide the references in the correct format (see line 336 and others).
We agree with the reviewers comment and made appropriate corrections.

Reviewer 3 Report
In this study, arginine (Arg) solution was used to surface treat PCL polymer film to enhance the MSC cell adhesion and spreading on the material. Arg concentration, temperature, and incubation time were investigated to examine their effects on the coating efficiency. After carefully evaluation, the reviewer recommends rejection of this manuscript. The main reasons are lack of novelty, missing essential data on material characterization, and unclear mechanisms underlying the chemistry. Other detailed comments are as follows:
- What interactions indeed cause the bonding of Arg to PCL? If there is chemical bonding involved, FTIR or NMR need to be performed to demonstrate the presence of –CO-NH- bond.
- Experiments need to be performed to characterize the surface charge of the prepared PCL films. Authors mentioned surface charge may determine PCL-Arg interaction and cell behavior. But there is only theory, no data at all. Did surface treatment really change the surface charge specifically in this system?
- Did heating change the mechanical and topographical properties of the PCL film? Only if authors rule out the potential changes of other variables brought by heating and soaking can we attribute the observed phenomenon to Arg concentration.
- What’s the topographical property of the prepared PCL film? Is it porous or smooth? Authors also need to show Arg present on the surface, not trapped in the PCL structures.
- How stable is this modified layer? Does Arg still stayed on the material surface after 3 days of cell culture?
- What is the potential application of this material? Although authors claimed in the last that “The presence of arginine influenced the degree of cellular adhesion, spreading, and proliferation”, the reviewer did not see significant improvement of material properties.
- If aminolysis reaction is claimed to be involved, control PCL should be treated with the solvent used for preparing arginine solutions, not simply unmodified PCL.
- Results from statistical analysis are not clearly presented. If there is significant difference, label in the graph and present the p-value. Since there are two critical variables, Arg concentration and temperature, two-way ANOVA should be used.
- Cell images (Figure 4C, D and E) are not quite consistent with the results presented in the graph (Figure 4I). According to the graph, Panel D is supposed to have the highest cell number while Panel E should have the lowest.
- Why cells in Figure 4B are significantly smaller (less spread) than those in Figure 5B? If they are cultured on the same substrate, are they supposed to look the same?
- Vinculin data is too simple. Area covered by vinculin clusters should be measured and quantitatively compared.
- Add scale bar information in the captions.
Author Response
In this study, arginine (Arg) solution was used to surface treat PCL polymer film to enhance the MSC cell adhesion and spreading on the material. Arg concentration, temperature, and incubation time were investigated to examine their effects on the coating efficiency. After carefully evaluation, the reviewer recommends rejection of this manuscript. The main reasons are lack of novelty, missing essential data on material characterization, and unclear mechanisms underlying the chemistry. Other detailed comments are as follows:
Response
Dear Reviewer 3, we are so grateful to you for review our manuscript and valuable comments. We took into account all your comments and recommendations. We supplemented manuscript with the results of studying the modified surface of a PCL film by FTIR and SEM methods. The resolution of the FTIR spectrometer made it possible to confirm the amide bond only in PCL film treated with 0.5 M arginine solution at heating. SEM results showed that treatment with arginine solution does not effect the surface topology of PCL films. The results were also supplemented by discussion and comparison with other published data.
- What interactions indeed cause the bonding of Arg to PCL? If there is chemical bonding involved, FTIR or NMR need to be performed to demonstrate the presence of –CO-NH- bond.
Surface molecular structure of modified PCL samples was analyzed using FTIR spectrometer. Figure 2 shows the FTIR spectra of PCL films before and after the arginine treatment. There are two amide bands resulting from nucleophilic attack of arginine on the carbonyl group of PCLs, which may be treated as evidence of arginine interaction with PCL surface. The first band in the range 1510–1580 cm-1 corresponds to amide II and associated with N–H bending vibrations. The second band in the range 1600–1700 cm-1 is assumed to amide I and associated with C=O (carbonyl) stretching vibration and C–N group vibrations. The resolving power of the FTIR spectrometer revealed the amide bond only in PCL samples treated with a 0.5 M arginine solution at 40áµ’C. The results of the ninhydrin analysis demonstrated that with such parameters of modification, most of all arginine binds to the surface. At room temperature, signals corresponding to the amide bond in the above ranges were not detected.
Figure 2. FTIR spectra of the various films: (A) unmodified PCL, (B) PCL treated with the 0,1M arginine solution for 24 hours at T=25C (C) PCL treated with the 0,5M arginine solution for 1 hour at T=40C. The two additional peaks at range from 1550 to 1650 cm-1 in aminolysed PCL films (C) indicated the introduction of amine groups onto the PCL substrates.
- Experiments need to be performed to characterize the surface charge of the prepared PCL films. Authors mentioned surface charge may determine PCL-Arg interaction and cell behavior. But there is only theory, no data at all. Did surface treatment really change the surface charge specifically in this system?
We agree with your comments and provided preliminary studies to evaluate the contact angle. However, only one experiment was realized and we can not give statistically reliable results.
- Did heating change the mechanical and topographical properties of the PCL film? Only if authors rule out the potential changes of other variables brought by heating and soaking can we attribute the observed phenomenon to Arg concentration.
New data was added. The changes in topography of the PCL film surfaces after modification were characterized by scanning electron microscopy (SEM). Fig. 3 shows the representative SEM images of the unmodified PCL and PCL films, treated with arginine solution. The initial state of PCL surface is a relatively smooth.. PCL is a crystalline polymer with a low melting point [Castilla-Cortázar, et al., 2019]. During the film fabrication, irregular hexagonal cells are formed. The size and distribution of these cells is shown in Fig. 4. The cells size does not exceed 100 µm. The SEM results do not revealed significant changes in the topology of the PCL surface after modification. Neither the concentration of arginine solution, processing time and temperature do not influence on the surface topology. The absence of pores on the PCL surface allows us to conclude that the physical sorption of arginine makes an insignificant contribution to the total amount of arginine on the modified film.
- Castilla-Cortázar, A. Vidaurre, B. Marí, A.J. Campillo-Fernández Morphology, Crystallinity, and Molecular Weight of Poly(ε-caprolactone)/Graphene Oxide Hybr. Pol. 2019, 11, 1099.
Figure 3. SEM images of PCL films; (A) unmodified PCL film; (B) film treated with the 0,1M arginine solution for 1 hour at T=40C; (C) film treated with the 0,25M arginine solution for 1 hour at T=40C; (D) film treated with the 0,5M arginine solution for 1 hour at T=40C; (E) film treated with the 0,1M arginine solution for 24 hours at T=25C (F); film treated with the 0,25M arginine solution for 24 hour at T=25C (G); treated with the 0,5M arginine solution for 24 hour at T=25C. Scale bar 100µm.
- What’s the topographical property of the prepared PCL film? Is it porous or smooth? Authors also need to show Arg present on the surface, not trapped in the PCL structures.
SEM results showed that the obtained PCL films have a smooth surface. The absence of pores on PCL surface allows us to make the assumption that the arginine physical sorption on PCL surface is unlikely.
- How stable is this modified layer? Does Arg still stayed on the material surface after 3 days of cell culture?
We realized additional studies to evaluate the interaction of arginine with the PCL surface. Modified films were incubated in water for 1 day. It was shown that the residual arginine content after 20 minutes and 24 hours incubation is no different. It should also be noted that the presence of amino groups on the surface is important only on the first day after cell seeding. In the process of cultivating cells from the culture medium, serum proteins are adsorbed onto the PCL film surface, as well as the extracellular matrix components synthesized by the cells, which cover the polymer surface not occupied by the cells.
- What is the potential application of this material? Although authors claimed in the last that “The presence of arginine influenced the degree of cellular adhesion, spreading, and proliferation”, the reviewer did not see significant improvement of material properties.
PCL is widely used in tissue engineering. One of the current areas of practical application of PCL is vascular surgery. However, the hydrophobicity and thrombogenicity of this material require its modification. At the moment, we are realized additional research the interaction of cells and their ability to synthesize extracellular matrix proteins. Depending on which proteins will preferably be synthesized by the cells on the modified surface, We can assumed what tissue can be regenerated by the modified PCL scaffold depend on extracellular matrix proteins synthesized by the cells on the modified PCL surface.
- If aminolysis reaction is claimed to be involved, control PCL should be treated with the solvent used for preparing arginine solutions, not simply unmodified PCL.
Our preliminary results have demonstrated that PCL surface treated with water without arginine does not differ from a PCL film without treatment. The cells interaction with such films also did not differ from cell adhesion and proliferation on unmodified films.
Fig. 2 hr cultivation: (A) - pure PCL,(B) - water treatment at 40C for 1 hr, (C) - water treatment at 25C for 24 hr. Scale bar 50 µm
1 day cultivation: (D) - pure PCL,(E) - water treatment at 40C for 1 hr, (F) - water treatment at 25C for 24 hr. Scale bar 50 µm
- Results from statistical analysis are not clearly presented. If there is significant difference, label in the graph and present the p-value. Since there are two critical variables, Arg concentration and temperature, two-way ANOVA should be used.
We agree with the comments of the reviewer and realized two-way statistical analysis. To analyze the statistically significant differences between specific samples the T-test was performed by using the Microsoft Excel Software. Samples were considered to be statistically important with the p<0,05.
- Cell images (Figure 4C, D and E) are not quite consistent with the results presented in the graph (Figure 4I). According to the graph, Panel D is supposed to have the highest cell number while Panel E should have the lowest.
We agree with the reviewer comments. The Fluorescence microscopy of the MSC data presented in Figure 5 demonstrate the cell spreading and the actin cytoskeleton organization. We estimated the number of adherent cells using the program ImageJ by the number of nuclei in the pictutes with a small magnification (10X). Analyzed samples are presented below.
Figure. Fluorescence microscopy of the MSC cells (DAPI) after 1 day of cultivation: (A) on the pure glass, (B) on the unmodified PCL, (C) on the film treated with the 0,1M arginine solution for 1 hour at T=40C, (D) on the film treated with the 0,25M arginine solution for 1 hour at T=40C, (E) on the film treated with the 0,5M arginine solution for 1 hour at T=40C, (F) on the film treated with the 0,1M arginine solution for 24 hours at T=25C, (G) on the film treated with the 0,25M arginine solution for 24 hour at T=25C, (H) on the film treated with the 0,5M arginine solution for 24 hour at T=25C. Magnification 10X.
- Why cells in Figure 4B are significantly smaller (less spread) than those in Figure 5B? If they are cultured on the same substrate, are they supposed to look the same?
We agree with the comments of the reviewer. Also in photographs that represent focal contacts (Fig. 7), the size of scale bar is larger compared to The size of scale bar in Figure 6 differs from scale bar in Figure 7. The fluorescence microscopy of cells on untreated PCL with the same scale bar present below.
- Vinculin data is too simple. Area covered by vinculin clusters should be measured and quantitatively compared
We analyzed the number of vinculin clusters using the program ImageJ according to the method proposed in the paper: U. Horzum, B. Ozdil, D. Pesen-Okvur Step-by-step quantitative analysis of focal adhesions MethodsX 1 (2014) 56–59
- Add scale bar information in the captions.
The authors agree with the comments and added a scale bar.

Round 2
Reviewer 2 Report
Despite authors not include mechanical characterization of the samples, I recommend to publish this submitted version of the manuscript.
Reviewer 3 Report
All the concerns are carefully addressed.